# Efficient Reprogramming of Mouse Embryonic Stem Cells into Trophoblast Stem-like Cells via Lats Kinase Inhibition

**DOI:** 10.3390/biology13020071

**Published:** 2024-01-24

**Authors:** Yake Gao, Wenrui Han, Rui Dong, Shu Wei, Lu Chen, Zhaolei Gu, Yiming Liu, Wei Guo, Fang Yan

**Affiliations:** 1State Key Laboratory of Conservation and Utilization of Bio-Resources, Center for Life Sciences, School of Life Sciences, Yunnan University, Kunming 650500, China; younkgao@mail.ynu.edu.cn (Y.G.); hanwenrui@mail.kiz.ac.cn (W.H.); gulei@mail.ynu.edu.cn (Z.G.);; 2Reproductive Medicine Center, Wuhan Women’s and Children’s Medical Care Center, Tongji Medical College, Huazhong University of Science and Technology, Wuhan 430030, China

**Keywords:** hippo pathway, lats kinase, reprogramming, embryonic stem cells, trophoblast stem cells

## Abstract

**Simple Summary:**

In this study, we investigated the critical process of cell fate determination in mouse embryos, focusing on the transition from embryonic stem cells (ESC) to trophoblast stem-like cells (TSLC). As ESCs and TSCs have distinct lineage differences and functional boundaries, the natural conversion between them is hindered. Through the inhibition of LATS kinase, we not only facilitated the conversion of inner cell mass (ICM) to trophectoderm (TE), but also successfully transformed ESC into stable self-renewing TSLC. Our findings highlighted the distinct molecular properties of TSLC, including the high expression of marker genes, such as *Cdx2* and *Eomes*, closely resembling those of trophoblast stem cells (TSC). Furthermore, TSLC demonstrated the ability to differentiate into mature trophoblast cells in vitro and actively participated in placenta formation in vivo.

**Abstract:**

Mouse zygotes undergo multiple rounds of cell division, resulting in the formation of preimplantation blastocysts comprising three lineages: trophectoderm (TE), epiblast (EPI), and primitive endoderm (PrE). Cell fate determination plays a crucial role in establishing a healthy pregnancy. The initial separation of lineages gives rise to TE and inner cell mass (ICM), from which trophoblast stem cells (TSC) and embryonic stem cells (ESC) can be derived in vitro. Studying lineage differentiation is greatly facilitated by the clear functional distinction between TSC and ESC. However, transitioning between these two types of cells naturally poses challenges. In this study, we demonstrate that inhibiting LATS kinase promotes the conversion of ICM to TE and also effectively reprograms ESC into stable, self-renewing TS-like cells (TSLC). Compared to TSC, TSLC exhibits similar molecular properties, including the high expression of marker genes such as *Cdx2*, *Eomes*, and *Tfap2c*, as well as hypomethylation of their promoters. Importantly, TSLC not only displays the ability to differentiate into mature trophoblast cells in vitro but also participates in placenta formation in vivo. These findings highlight the efficient reprogramming of ESCs into TSLCs using a small molecular inducer, which provides a new reference for understanding the regulatory network between ESCs and TSCs.

## 1. Introduction

Cell fate determination is a crucial process in early mammalian embryos and plays a pivotal role in successful embryo implantation and the maintenance of pregnancy. In mice, this process occurs in two stages of lineage differentiation before implantation. The first stage, known as embryonic compaction (16–32 cells), gives rise to two lineages: trophectoderm (TE) and inner cell mass (ICM) [1,2,3,4,5]. The TE lineage, which is responsible for embryo implantation and placenta formation [3,5,6], is formed during this stage. Following this, the embryo progresses to the blastocyst stage, where the ICM further differentiates into the pluripotent Epiblast (EPI) lineage and the primitive endoderm (PrE) lineage [7,8,9]. The EPI lineage develops into the fetal body, while the PrE lineage forms the yolk sac [10,11,12]. To study these lineages and understand the molecular mechanisms underlying lineage differentiation, researchers have successfully derived trophoblast stem cell lines (TSC), embryonic stem cell lines (ESC), and extra-embryonic endoderm stem cell lines (XENC) [13,14,15]. These in vitro models provide important tools for further investigations.

Differential activation of the HIPPO signaling pathway plays a critical role in determining the fate of the first cells in the mouse embryo. This process begins at the 8-cell stage when the inner and outer blastomeres undergo distinct changes [13,16]. One important aspect of this decision-making process is the polarization of the blastomeres along the apical-basal axis. This involves the uneven distribution of membrane proteins and cytoskeletal components in the apical and basolateral domains, the establishment of an apical domain, actin, and apical actomyosin contraction leading to embryo compaction [17,18,19]. As a result of this polarization, the 8-cell blastomeres divide in different directions relative to the embryonic axis, giving rise to inner and outer cells with distinct characteristics [7,18,20,21]. By the 16-cell stage, these populations of cells begin to emerge, with inner cells being polar and establishing contacts with surrounding cells, while outer cells exhibit polarization along the radial axis of the embryo and lack cell contacts on the outer surface [2,22,23,24]. This mode of cell polarity is a major pathway that regulates HIPPO signaling activity [16,22,25,26]. In outer cells, the establishment of this polarity prevents the LATS1/2 kinase from phosphorylating the transcriptional coactivator YAP, allowing YAP to accumulate in the nucleus [9,27,28]. On the other hand, in inner cells, phosphorylated YAP is sequestered in the cytoplasm [5,28]. Recent live dynamic imaging studies have shown that in 16–32-cell stage embryos, inner cells exhibit extranuclear localization of YAP, while externally polarized cells display intranuclear localization of YAP [29,30]. The binding of nuclear YAP to TEAD4 is necessary for the formation of the TE and plays a critical role in activating downstream genes such as *Cdx2* and *Gata3* [24,26,27]. CDX2, a TE-specific transcription factor, is activated upon the binding of YAP/TEAD4 to its promoter, leading to the downregulation of *Oct4* and *Nanog*, which are specific to the inner cell mass (ICM) [16,30,31,32]. Consistent with this, overexpression of LATS1/2 kinase reduces CDX2 expression in outer cells, while inhibiting LATS1/2 in embryos results in CDX2 expression in inner cells, thereby downregulating ICM-specific gene expression [23,33,34,35]. In summary, during early embryonic development, the activation of HIPPO signaling plays a crucial role in determining the fate of the first cells. The establishment of blastomere polarization and the subsequent differential localization of YAP contribute to the formation of inner and outer cells with distinct characteristics, ultimately leading to the differentiation of the trophectoderm and the downregulation of inner cell mass-specific genes.

Embryonic stem cells (ESCs) are pluripotent stem cells derived from preimplantation embryos, capable of developing into all cell types of the fetus. They offer a new approach for generating gene-edited mice [36,37,38,39]. ESCs are also valuable for studying lineage differentiation in implantation embryos, creating models of blastocyst-like structures (blastoids) and gastrulating embryo-like structures (gastruloids), and simulating the morphological differentiation of post-implantation embryos [40,41,42,43,44,45]. ESCs, derived from the inner cell mass of the embryo, resemble the preimplantation epiblast and are referred to as naive state cells [46,47]. Subsequently, epiblast stem cells (EpiSCs), resembling cells of the anterior primitive streak, are established from post-implantation mouse embryos. They co-express pluripotency and lineage-specific genes and are considered primed state cells [48,49,50]. Additionally, there is a transient state of pluripotency known as the formative state cells, which corresponds to the early post-implantation epiblast and represents a state between naive ESCs and EpiSCs [51,52]. The current focus of mouse embryonic stem cell research is primarily on naive ESCs to explore their full lineage differentiation potential. Trophoblast stem cells (TSCs) are derived from the polar trophoblast of the blastocyst or the extraembryonic ectoderm and have the ability to differentiate into all trophoblast subtypes [14,53,54,55]. Fibroblast growth factor 4 (FGF4) induces ERK phosphorylation in some polar trophoblast cells, promoting TSC proliferation and maintaining their undifferentiated state [14,56,57,58]. Removal of FGF4 results in a significant reduction in the expression of undifferentiated marker genes, such as *Cdx2*, and a halt in cell proliferation [59,60]. Studies have revealed that TSCs exhibit heterogeneity, similar to the implantation stage of the embryo, consisting of various fluctuating and interchangeable subtypes of cells [61,62,63,64,65]. However, by simulating and optimizing the combination of inducers, stable trophectoderm stem cells (TESCs) can be generated. These TESCs possess high self-renewal capacity, inhibit premature differentiation, and exhibit better trophectoderm differentiation potential, similar to the blastocyst stage [61,66].

Mouse embryonic stem cells lack the ability to differentiate into TE cells due to their origin from the inner cell mass (ICM) or epiblast (EPI), which do not contribute to TE lineages in chimeric blastocysts [36,67]. However, down-regulation of the pluripotency gene *Oct4* in ESCs can induce their differentiation toward TE cells, leading to the formation of multinuclear giant cells [68]. The addition of FGF4 to OCT4 down-regulated ESCs enables them to acquire molecular characteristics resembling trophoblast stem cells (TSC) and continue proliferating [14,31]. Furthermore, activation of CDX2 can trigger TE differentiation, including TSC derivation, suggesting its critical role in TE cell fate determination [31,36]. It has been observed that OCT4 and CDX2 establish and maintain their individual expression patterns through a mutual inhibitory loop [31,68,69], and their transformation between stable and superior expression patterns forms a barrier [31,36]. TEAD4, a transcriptional effector of the HIPPO signal transduction system, plays a key role in initiating TE lineage differentiation [3,24,27]. Overexpression of TEAD4 can up-regulate the expression of CDX2, inducing ESCs to differentiate into TE cells [3,27]. Moreover, the overexpression of TSC-specific transcription factors *Cdx2*, *Gata3*, *Arid3a*, *Tfap2c*, and *Esrrb* enables the reprogramming of ESCs or fibroblasts into TS-like cells [35,70,71,72,73,74,75]. However, these genetic manipulation methods have high technical requirements. In this study, we successfully reprogrammed mouse ESCs into TSC-like cells (TSLCs) with stable self-renewal by inhibiting LATS kinase, facilitating YAP binding to TEAD4, and up-regulating the expression of TE core genes such as *Cdx2* and *Eomes* [76,77]. TSLCs exhibit similar molecular properties to TSCs, including high expression of *Cdx2*, *Eomes*, and *Tfap2c*, as well as hypomethylation of their promoter regions. Importantly, TSLCs not only possess the ability to differentiate into mature trophoblast cells in vitro but can also participate in placenta formation in vivo.

## 2. Methods and Materials

### 2.1. Animal Care and Use

The ICR and B6-H11-CAG-EGFP mice were procured from CARR, Yunnan University. These mice were housed at CARR under controlled conditions with a temperature maintained between 22–24 °C and a light/dark cycle of 12 h each. They had continuous access to food and water.

### 2.2. Mouse Embryo Culture

Female mice, aged 8–10 weeks, were administered an intraperitoneal injection of 8 IU of PMSG (Prospec, Ness Ziona, Israel, HOR-272), followed by a subsequent injection of 8 IU of HCG (Macklin, Shanghai, China, C805163) 48 h later. These female mice were then paired in a 1:1 ratio with male mice of normal fertility, and vaginal plugs were checked the following morning to confirm successful mating. In addition, 2-cell embryos or blastocysts were obtained by flushing the fallopian tubes or uteri of the female mice. These embryos were cultured in G-1^TM^ (Vitrolife, Gothenburg, Sweden, 10128) or G-2^TM^ (Vitrolife, 10132) medium, which was covered with mineral oil (Vitrolife, 10029). The cultures were maintained under the following conditions: 37 °C temperature, 6% CO_2_, and 5% O_2_. G-1^TM^ medium was used for embryos at the pre-16-cell stage, while G-2^TM^ medium was used for embryos at the 16-cell stage or beyond.

### 2.3. Stem Cells and Culture Conditions

Mouse ESC lines derived from E3.5 blastocysts (B6-H11-CAG-EGFP or ICR) were cultured using the methodology described by Ying et al. [78]. The ESC medium consisted of a N2B27-based medium supplemented with either 10 ng/mL mLIF (Merck, Darmstadt, Germany, ESG1107) or hLIF (PeproTech, Cranbury, NJ, USA, 300-05), 1 μM PD0325901 (Tocris, Bristol, UK, 4192/10), and 3 μM CHIR99021 (Tocris, 4423/10). The N2B27 basal medium was prepared by mixing DMEM/F-12 (Gibco, Waltham, MA, USA, 10565018) and Neurobasal (Gibco, 21103049) in a 1:1 ratio. To this mixture, 100X N2 additive (17502-048), 100X B27 additive (17504-044), 0.1 mM 2-mercaptoethanol (Gibco, 31350010), 100X MEM NEAA (Gibco, 2281490), 100X GlutaMAX (35050038), and 5% KSR (Gibco, 10828-028) were added. ESCs were maintained on MEF feeder cells and passaged every 3 days with a 1:10 dilution.

The mouse TSC line used in this study was originally gifted by Dr. Janet Rossant’s laboratory at The Hospital for Sick Children, University of Toronto. The TSC medium consisted of RPMI-1640 (Gibco, 61870036) supplemented with 20% FBS (VivaCell, Denzlingen, Germany, C04002-500), 0.1 mM 2-mercaptoethanol (Gibco, 31350010), 100X Sodium Pyruvate (Gibco, 11360070), 25 ng/mL rhFGF4 (R&D system, Minneapolis, MN, USA, 235-F4), and 1 mg/mL heparin (Sigma, Darmstadt, Germany, H3149). The TSCs were cultured on feeder cells (MEF) and were passaged every 3 days with a 1:6 dilution.

Reprogramming of ESCs into TSLCs: ESCs were cultured on feeder cells with ESC medium, upon reaching 50–60% confluence, ESC medium was substituted with TSC medium supplemented with 3 μM LATS-IN-1 (TargetMol, Boston, MA, USA, T9053) to reprogram ESCs into TSC-like cells. The induction medium (+LATS-IN-1) was replaced every day, and passage culture was performed when cells reached 80–90% confluence with a 1:4 dilution. Cell dissociation was performed using either TrypLE™ Express (Gibco, 12605010) or StemPro™ Accutase™ (Gibco, A1110501). At each passage, whole cells were dissociated and seeded on feeder cells without clone selection, 5 μM Y27632 (Tocris, Bristol, UK, 1254) was added to improve cell attachment, with subsequent removal the following day, each passage cycle was 3–4 days. About 15 days after induction, ESCs successfully underwent complete reprogramming into TSLCs that could be sustained with TSC medium. TSLCs were cultured on feeder cells and were passaged for about 3–4 days with a 1:4–1:6 dilution.

### 2.4. Immunofluorescence of Embryos or Cells

Samples were immersed in a 4% paraformaldehyde solution for 15–20 min at room temperature. Afterward, they were washed three times with PBS for 5 min each time. Permeabilization was achieved by treating the samples with a 0.25% solution of Triton X-100 in PBS (PBS-T) for 30 min at room temperature. To prevent non-specific binding, a blocking solution consisting of a 4% bovine serum albumin (BSA) solution in 0.1% PBS-T was applied. The samples were then incubated overnight at 4 °C with the primary antibody (diluted in the blocking solution). Detailed information about the antibodies used in this study can be found in Appendix A. Subsequently, the samples underwent three washes with PBS containing 0.1% Tween-20 for 5 min each, followed by a 2-h incubation at room temperature with the appropriate secondary antibodies. Nuclear staining was performed using DAPI. Imaging was performed using ZEISS Axiovert 200 microscopy (ZEISS, Jena, Germany) or ZEISS LSM 880 Confocal microscopy (ZEISS, Jena, Germany), and subsequent analysis was conducted using ZEN Blue (Black) Lite2_3 and ImageJ_V1.8.0 software.

### 2.5. Quantitative RT-PCR

The Cell Total RNA Isolation Kit (Foregene, RE-03111, Chengdu, China) was used to extract RNA. Subsequently, cDNA synthesis was carried out using the EasyScript^®^ cDNA Synthesis SuperMix Kit (Transgene, AE341-02, Beijing, China). The RT-qPCR analysis was performed on a Bio-Rad CFX96TM Real-Time PCR System using SsoFast EvaGreen^®^ SuperMix (BIO-RAD, Hercules, CA, USA, 1725201) and Hard-Shell PCR 96-well Plates (BIO-RAD, HSP9655). The relative mRNA expression levels were determined using the 2^−ΔΔCT^ method with *Gapdh* as the reference gene. Detailed primer sequences can be found in Appendix A.

### 2.6. Flow Cytometry

Cells were dissociated into individual cells using a 0.25% trypsin solution for 5 min, followed by fixation in a 4% polyformaldehyde solution for 15 min. Permeabilization was achieved by incubating the cells in a 0.25% Triton X-100 solution in PBS (PBS-T) for 15 min. After that, the cells were treated with a 4% BSA (Sigma, A1933) solution in 0.1% PBS-T for 1 h and then incubated with primary antibodies (CDX2 and GATA6) for 2 h. Subsequently, the cells were incubated with appropriate secondary antibodies at room temperature for 1 h. After incubation, the cells were washed, resuspended in FACS buffer (PBS + 2% FBS), and analyzed using a BD Aria SORP cell sorter. A minimum of 10,000 cells were used for gating in each assay, and Flow Jo V10.8.1 software was used for data analysis. The fluorophores used with their corresponding gating channels were Alexa 488 for FITC, Alexa 568 for PE, and Alexa 647 for APC. All procedures were performed at room temperature.

### 2.7. Chimeric Embryos and Implantation

Trophoblast stem-like cells (TSLCs) that expressed green fluorescent protein (GFP) were isolated and kept at a temperature of 4 °C. Afterward, a microinjection system (TransferMan-4r, Eppendorf) was used to inject 6–8 individual TSLCs into the cavity of a blastocyst at embryonic day 3.5. The embryos were then cultured in a G-2 medium for an additional 12 h before being transferred into the uterus of recipient animals. To assess the occurrence of successful implantations, the conceptus was gently exfoliated in PBS 10 days post-embryo transfer.

### 2.8. Quantification and Statistical Analysis

Statistical analyses were conducted using GraphPad Prism 8 (GraphPad Software) and SPSS Statistics 26.0 (IBM, Armonk, NY, USA). Mean ± SEM was used to represent the error bars, and the number of biological replicates for each experiment was indicated in the figure legends. The student’s *t*-test was used for statistical analyses, with a *p*-value threshold of <0.05 considered as statistically significant for all cases.

## 3. Results

### 3.1. The Cellular Localization of YAP Regulates the Differentiation State of TSCs

The localization of YAP within cells plays a crucial role in the initiation and regulation of TE differentiation in preimplantation embryos, and this process is dependent on the activity of LATS kinase [34,35]. To investigate the involvement of HIPPO signaling in trophoblast stem cells (TSCs), we targeted LATS kinase and inhibited its activity using a small molecule compound called LATS-IN-1 [79]. First, we validated the effectiveness of the LATS kinase inhibitor in promoting the nuclear accumulation of YAP in mouse embryonic fibroblasts (Appendix A). In TSCs, the use of LATS inhibitors led to increased CDX2 expression and YAP accumulation in the nucleus (Figure 1A and Appendix A). As observed in previous studies, control TSCs exhibited heterogeneity with uneven CDX2 expression and YAP accumulation both inside and outside the nucleus (Figure 1A and Appendix A). After three days of random differentiation (FGF4 removal), TSCs showed minimal CDX2 expression, while YAP predominantly accumulated in the cytoplasm (Figure 1A and Appendix A). This observation was verified using RT-qPCR, which demonstrated significant upregulation of marker genes (*Cdx2*, *Elf5*, *Eomes*, *Tfap2c*, *Gata3*, and *Gata2*) in TSCs treated with the inhibitor compared to untreated TSCs (Figure 1B). Conversely, the expression of differentiation genes (*Gcm1*, *Ctsq*, *Tpbpa*, *Prl3d1*, *SynA*, and *Ascl2*) was significantly reduced in TSCs after inhibitor treatment (Figure 1C). Importantly, in differentiated TSCs, the expression of these genes exhibited opposite trends (Figure 1B,C), suggesting that the localization of YAP within cells is involved in maintaining stemness and driving differentiation in TSCs. Furthermore, we assessed whether the LATS inhibitor could reverse the differentiation of TSCs by treating differentiated TSCs with the inhibitor. Interestingly, the inhibitor altered the cellular localization of YAP in differentiated TSCs, leading to increased nuclear accumulation (Figure 1D and Appendix A). This effect was accompanied by significant upregulation of TSC marker genes (*Cdx2*, *Elf5*, *Eomes*, and *Tfap2c*) and downregulation of differentiation genes (*Gcm1*, *Ctsq*, and *Prl3d1*) (Figure 1E). These findings suggest that the cellular localization of YAP actively regulates the differentiation state of TSCs. In summary, by inhibiting LATS kinase activity, we observed changes in YAP localization and gene expression patterns that indicate a shift towards stemness or differentiation in TSCs.

### 3.2. Inhibition of LATS Kinase Promotes the Specialization of Embryos into the TE Lineage

We further investigated the impact of LATS kinase inhibitors on embryonic lineage differentiation. We observed that at the 8-cell stage, CDX2 expression was minimal in embryos (Figure 2A). However, when embryos were treated with LATS-IN-1, we observed an increase in CDX2 expression and a significant rise in the number of CDX2 positive cells (Figure 2B,C and Appendix A). This increase in CDX2 expression and the upregulation of *Cdx2*, *Tead4*, and *Eomes* mRNA levels (Figure 2G and Appendix A) indicate that inhibiting LATS kinase might prematurely activate TE lineage differentiation. Moving on to the 16-cell stage, CDX2 was predominantly expressed in the outer blastomeres. However, in treated embryos, we also observed CDX2 expression in the internal blastomeres, resulting in a significant increase in the number of CDX2 positive cells (Figure 2A,B,D and Appendix A). Conversely, the number of OCT4 positive cells was significantly reduced, and OCT4 expression was attenuated (Figure 2A,B,D and Appendix A). As expected, there was a significant increase in the expression of *Cdx2*, *Tead4*, and *Eomes*, while *Oct4* expression was significantly decreased in treated embryos (Figure 2H and Appendix A). At the 32-cell stage, the expression pattern of ICM and TE marker genes in treated embryos resembled that of the 16-cell stage. We observed a significant increase in the number of CDX2 positive cells and a significant decrease in the number of OCT4 positive cells compared to control embryos (Figure 2A,B,E, and Appendix A). Consistent with these findings, the expression levels of *Cdx2*, *Tead4*, and *Eomes* were significantly elevated, while *Oct4* and *Sox2* expression showed a significant decrease (Figure 2I and Appendix A). By the blastocyst stage (64-cell), the treated embryos exhibited no significant OCT4 expression, but CDX2 was expressed in almost all blastomeres (Figure 2B,F and Appendix A). Furthermore, compared to the control group, the expression of *Cdx2*, *Tead4*, and *Eomes* was significantly increased, while *Oct4* and *Sox2* expression was significantly decreased in the treated embryos (Figure 2J and Appendix A). Additionally, we noticed a substantial accumulation of YAP1 in the nucleus of blastomeres from LATS-IN-1 treated blastocysts (E4.0), whereas, in the control group, less YAP1 accumulated in the nucleus (Figure 2K and Appendix A). Thus, we hypothesized that inhibiting LATS kinase could alter the localization of YAP in blastomeres, enhance the binding of YAP to TEAD4, promote TE specialization, and impede the lineage segregation of ICM-TE.

### 3.3. Inhibition of LATS Kinase Promotes ESCs Transformation to TSC-like Cells

Inhibition of LATS kinase can result in increased nuclear localization of YAP, leading to the binding of YAP and TEAD4 and initiating the expression of TE core genes. With this in mind, we conducted an experiment to investigate whether LATS kinase inhibition can break down the differentiation barrier between trophoblast stem cells (TSCs) and embryonic stem cells (ESCs). In this experiment, ESCs were induced on feeder cells using two different culture mediums: classic TSC medium (control) and TSC medium supplemented with 3 µM LATS-IN-1 (LATS-IN). The results showed that after 10–15 days of induction culture, ESCs in the control group exhibited very low expression of CDX2, with only 6.59% of cells expressing CDX2 (Figure 3A,D). On the other hand, a high percentage (92.2%) of cells in the control group expressed GATA6 (Figure 3A,D). This suggests that the TSC medium is unable to induce the conversion of ESCs into TSCs but instead generates XENC-like cells. In contrast, ESCs in the inhibitor group showed a high expression of CDX2, with 73.1% of cells expressing CDX2 (Figure 3A,D). However, the expression of GATA6 was almost absent, with only 5.13% of cells expressing GATA6 (Figure 3A,C,D). The results from RT-qPCR were consistent with these findings. In the control group, the expression of primitive endoderm (PrE) marker genes *Gata6*, *Gata4*, and *Pdgfra* significantly increased with induction time, while the expression of TSC marker genes *Cdx2*, *Eomes*, *Elf5*, *Tfap2a*, *Hand1*, and *Gata3* showed a transient increase followed by a rapid decrease (Figure 3E,F). Similarly, in the inhibitor-treated ESCs, the expression pattern of *Gata6*, *Gata4*, and *Pdgfra* resembled that of TSC marker genes in the control group, with a transient increase followed by a rapid decrease (Figure 3G). However, the expression of *Cdx2*, *Eomes*, *Elf5*, *Tfap2a*, *Hand1*, and *Gata3* was significantly increased (Figure 3H). Based on these results, we conclude that the addition of LATS kinase inhibitors to the TSC medium can reprogram ESCs into TSC-like cells. Furthermore, we observed a positive correlation between the expression of CDX2 and the accumulation of YAP1 in the nucleus in TSC-like cells (Figure 3B and Appendix A). We found that after 8 days of inhibitor induction, there were still a small number of OCT4+ cells (Figure 3C). Although the expression of OCT4 decreased overall, CDX2+ cells and OCT4+ cells co-existed at this time, which was a mixed state (Figure 3C and Appendix A). Interestingly, we also found that the percentage of CDX2 positive cells increased while the percentage of KRT7 positive cells decreased with increasing induction culture time in TSC-like cells (Appendix A). Based on this, we speculate that KRT7 may serve as a transition marker for the transformation of ESCs into TSC-like cells (Appendix A), and the transformation of ESCs into TSC-like cells is a progressive process.

### 3.4. TSLCs Can Differentiate into Placental Terminal Trophoblast Cells In Vivo

In addition, we compared the molecular properties of Trophoblast Stem-like Cells (TSLCs) with Trophoblast Stem Cells (TSCs). Similar to TSCs, TSLCs did not express OCT4, which is known as a marker gene for Embryonic Stem Cells (ESCs). However, they highly expressed CDX2, a marker gene for TSCs (Figure 4A). RT-qPCR results demonstrated that the expression of TSC marker genes, such as *Cdx2*, *Elf5*, *Eomes*, *Tfap2c*, and *Gata2*, in TSLCs was similar to that in TSCs (Figure 4C). This similarity was further confirmed by bisulfite sequencing, which showed that the promoter region of *Oct4* was hypomethylated in ESCs but hypermethylated in both TSCs and TSLCs (Figure 4B). In contrast, the promoter regions of *Cdx2* and *Tead4* were hypermethylated in ESCs but hypomethylated in TSCs and TSLCs (Figure 4B). Similar patterns of methylation were observed in the promoter regions of *Tfap2c* and *Krt7* compared to ESCs and TSCs (Appendix A). These findings collectively indicate that TSLCs and TSCs share similar molecular properties. Since TSCs have been shown to play a role in placental development in chimeric embryos [14], we conducted chimeric embryo experiments on TSLCs to investigate their biological functions. To achieve this, we reprogrammed ESCs (GFP labeled) to TSLCs (GFP labeled) (Figure 4D) and injected 6–8 TSLCs (GFP labeled) into E3.5 blastocysts (Figure 4E). These blastocysts were then transferred into the recipient’s uterus 12 h later. The results demonstrated that TSLCs contributed to the formation of the E13.5 placenta in the recipient (Figure 4F). Furthermore, the expression of the mature trophoblast cell-specific protein TPBPA and the syncytiotrophoblast cell marker SYNA were both co-located with TSLCs (GFP-positive) in the deuterogenic cells (Figure 4G and Appendix A). These findings provide evidence that TSLCs have the ability to differentiate into placental terminal trophoblast cells in vivo.

### 3.5. TSLCs Can Differentiate into Mature Trophoblast Cells In Vitro

Fibroblast growth factor 4 (FGF4) plays a crucial role in maintaining the self-renewal of trophoblast stem cells (TSCs) in vitro (Figure 5A) [14,59,80]. In the absence of FGF4, TSCs randomly differentiate into various trophoblast cell subtypes in vitro, including trophoblast giant cells (TGCs), parietal TGCs (P-TGCs), sinusoidal TGCs (S-TGCs), spiral associated TGCs (SpA-TGCs), canal TGCs (C-TGCs), spongiotrophoblasts (SpTs), syncytiotrophoblast layer I (SynT-I), and syncytiotrophoblast layer II (SynT-II) [59,60,81,82]. To investigate the differentiation potential of trophoblast-specific lineage cells (TSLCs), we cultured them in a TSC medium without FGF4. As the differentiation time increased, the expression of TSC marker genes *Cdx2* and *Eomes* significantly decreased (Figure 5B). Conversely, the expression of TGC marker genes *Ctsq* and *Prl3d1* significantly increased (Figure 5C,D), along with the expression of SPT marker gene *Tpbpa* (Figure 5E). Additionally, the expression of SynT markers *Syna* and *Synb* showed a significant increase (Figure 5F,G). These findings were consistent with the protein expression of major mature trophoblast cell markers KRT7, TPBPA, and SYNA in TSLCs after 4 days of differentiation culture, while the expression of TSC marker TFAP2C was significantly decreased (Figure 5H and Appendix A). In summary, our results demonstrate that TSLCs have the potential to differentiate into mature trophoblast cells both in vivo and in vitro. Inhibition of LATS kinase serves as an effective induction strategy for reprogramming ESCs into TSLCs.

## 4. Discussion

Embryonic stem cell lines derived from early embryos are crucial for studying lineage differentiation and expanding gene editing techniques, particularly for verifying the full lineage development potential of ESCs through tetraploid compensation [39,83,84]. Trophoblast stem cell lines serve as an essential model for understanding embryo implantation and placental differentiation. However, there are still many unknown regulatory mechanisms and developmental potential of trophoblast stem cells [85]. Functional verification similar to ESC tetraploid compensation, whether trophoblast stem cells can completely replace TE function, is not yet to be achieved. Moreover, the expanded potential stem cells (EPSCs) [11,12] have shown extraembryonic differentiation potential, while the ability to differentiate into TE is limited [86]. Previous studies have shown that both morula-like cells (MLCs) [87] and 2C-like cells [88,89,90] have the potential to differentiate towards TE. It may be that they capture the molecular properties of earlier embryonic cells with higher developmental potential. In order to realize blastoid implantation and sustained development, it is imperative to further investigate TSC differentiation. We have discovered that the heterogeneity of TSCs is linked to varying degrees of cytoplasmic accumulation of YAP (Figure 1A). Increased cytoplasmic accumulation of YAP promotes TSC differentiation and down-regulates the expression of core self-renewal genes such as *Cdx2* and *Eomes* (Figure 1B,C), similar to Mural trophectoderm (MTE) [91]. Inhibition of LATS kinase significantly increases YAP accumulation in the nucleus, preventing TSC differentiation and improving TSC marker gene expression (Figure 1A,C), similar to Polar trophectoderm (PTE) [61]. However, it remains unclear if this relatively homogeneous high CDX2 expression aligns with the TE differentiation state (Figure 2K and Appendix A) or if it expands the differentiation potential of TSCs.

YAP localization plays a crucial role in regulating the differentiation of TE and inner cell mass (ICM) lineages during early embryonic development [3,24,35]. Previous studies have shown that YAP is primarily localized in the nucleus of outer blastomeres in the late 8-cell stage to the 32-cell stage, while in ICM cells, YAP is predominantly found in the cytoplasm [29]. However, the dynamics of YAP localization in blastocyst TE during the further differentiation into mural trophectoderm (MTE) and polar trophectoderm (PTE) remain unclear. Furthermore, we investigated the effects of adding a LATS inhibitor during the reprogramming of embryonic stem cells (ESCs). We observed that this manipulation led to a distinct cell fate compared to control conditions. Under trophoblast stem cell (TSC) culture conditions, ESCs treated with the LATS inhibitor transdifferentiated into extraembryonic endoderm (XENC)-like cells expressing *Gata6*, along with a transient upregulation of TSC marker genes (Figure 3F). This change in cell fate may be attributed to the deregulation of transcriptional control of TE genes, such as *Cdx2*, following the downregulation of *Oct4* [31,68]. In contrast, when ESCs were stimulated with FGF4, the activation of FGFR receptors led to the expression of primitive endoderm (PrE) lineage genes such as *Gata6* and *Gata4* [7,58], promoting the conversion of ESCs into XENC-like cells. However, in the presence of LATS inhibitors, although similar control signals were present, there was an increase in nuclear YAP localization, resulting in the upregulation of TEAD-dependent TE lineage genes such as *Cdx2*, *Eomes*, and *Gata3* (Figure 3H). This finding aligns with the mechanism underlying the reprogramming of ESCs into TS-like cells via overexpression of these genes [70,71,73,75]. This sustained increase in the expression of core TE genes inhibited the transformation of ESCs into PrE cells, leading to only a transient elevation of *Gata6* and *Gata4* expression (Figure 3G).

Although TSLCs and TSCs share similar molecular properties, including high expression of core gene *Cdx2*, *Eomes*, *Elf5*, and *Tfap2c* (Figure 4C), and hypomethylation of *Cdx2*, *Tead4*, and *Tfap2c* promoter regions (Figure 4B), we still cannot determine whether these two can somehow replace each other. We did not further analyze whether TSLCs differ from TSCs in epigenetics and proteomics. We demonstrated that TSLCs have the same differentiation potential as TSCs, including contribution to the formation of recipient placenta and differentiating into mature trophoblast cells in vitro. TSLCs were not involved in the formation of the fetus and yolk sac of the recipient (Figure 4F), indicating that the induced ESCs had lost pluripotency and had been reprogrammed into distinct TE lineage cells. We found that lacking LIF caused a decrease in the proliferation of ESCs during induction culture, especially during the prophase of induction culture. Meanwhile, Y27632 (a Rho-ROCK inhibitor) at a moderate concentration contributed to the attachment and proliferation of ESCs at the passage of induction culture (results not shown). Interestingly, we found that TSCs differentiation marker KRT7 was early expressed and co-localized with CDX2, but gradually lost expression as CDX2 expression increased during ESCs induction culture (Appendix A). Therefore, KRT7 may be a transition marker of the transformation of ESCs into TSCs. Ultimately, the addition of LATS inhibitors resulted in the reprogramming of ESCs into TSC-like cells. Overall, our findings highlight the critical role of YAP localization in determining the fate of TE and ICM lineages during early embryonic lineage differentiation. Additionally, we demonstrate that manipulating YAP activity using LATS inhibitors can drive the generation of distinct cell populations with implications for embryonic development and stem cell biology.

## 5. Conclusions

Our findings demonstrate that inhibiting LATS enhances the nuclear localization of YAP, resulting in increased expression of CDX2 in TSCs, thereby impeding their differentiation potential. Inhibition of LATS also leads to diminished expression of the ICM marker, OCT4, and enhanced expression of the TE marker, CDX2, in early embryos. Consequently, this impedes the segregation of ICM-TE lineages and promotes the transformation of ICM into TE lineage. Remarkably, while the TSC medium fails to reprogram ESCs into TSCs, the addition of the LATS inhibitor LATS-IN-1 effectively reprograms them into TSLCs. TSLCs exhibit both molecular similarities to TSCs and the ability to differentiate into mature trophoblast cells both in vivo and in vitro. Thus, inhibition of LATS presents a compelling strategy for facilitating the transformation of ESCs into TSCs.

## Figures and Tables

**Figure 1 biology-13-00071-f001:**
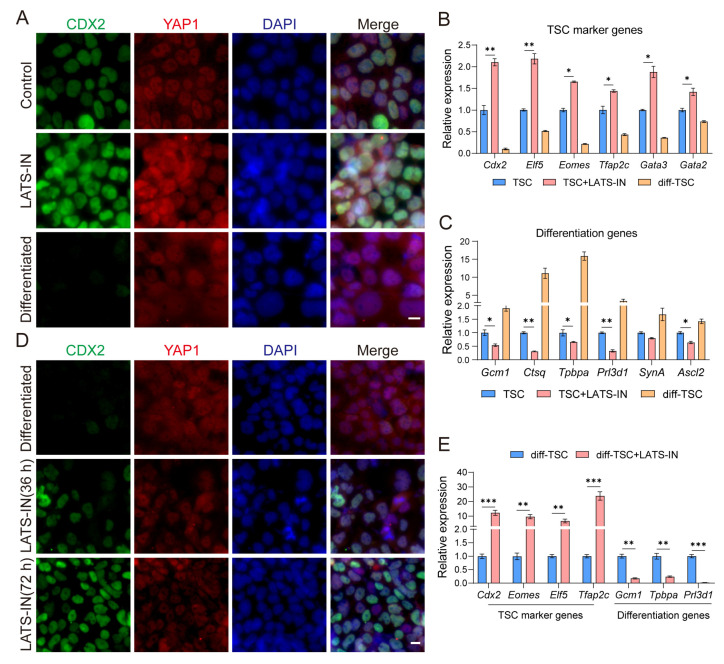
The cellular localization of YAP regulates self-renewal and differentiation of trophoblast stem cells. (**A**) Immunostaining for CDX2 and YAP1 of different types of trophoblast stem cells. “Control” refers to the conventional TSC culture medium; “LATS-IN” refers to the addition of a 3 µM concentration of LATS-IN-1 (An inhibitor of Lats kinase) to the TSC culture medium; “Differentiated” refers to the TSC after 3 days of random differentiation. Cell nuclei were stained with DAPI (blue). (**B**) Relative expression of *Cdx2*, *Elf5*, *Tfap2c*, *Eomes*, *Gata3*, and *Gata2* in TSC, LATS-IN-1 treated TSC, and differentiated TSC. (**C**) Relative expression of *Gcm1*, *Ctsq*, *Tpbpa*, *Prl3d1*, *SynA*, and *Ascl2* in TSC, LATS-IN-1 treated TSC, and differentiated TSC. (**D**) Immunostaining for CDX2 and YAP1 of differentiated TSC and differentiated TSC was treated with LATS-IN-1 factor for 36 h and 72 h, the instructions are the same as (**A**). Cell nuclei were stained with DAPI (blue). (**E**) Relative expression for *Cdx2*, *Elf5*, *Tfap2c*, *Eomes*, *Gcm1*, *Tpbpa*, and *Prl3d1* in differentiated TSC and LATS-IN-1 treated differentiated TSC. All gene relative expression was detected by qRT-PCR, and the data were normalized to *Gapdh*, *n* = 3. For each graph, the data were represented as mean ± SEM, analyzed by Student’s *t*-test, * *p* < 0.05, ** *p* < 0.01, *** *p* < 0.001. Scale bars: 10 μm in (**A**,**D**).

**Figure 2 biology-13-00071-f002:**
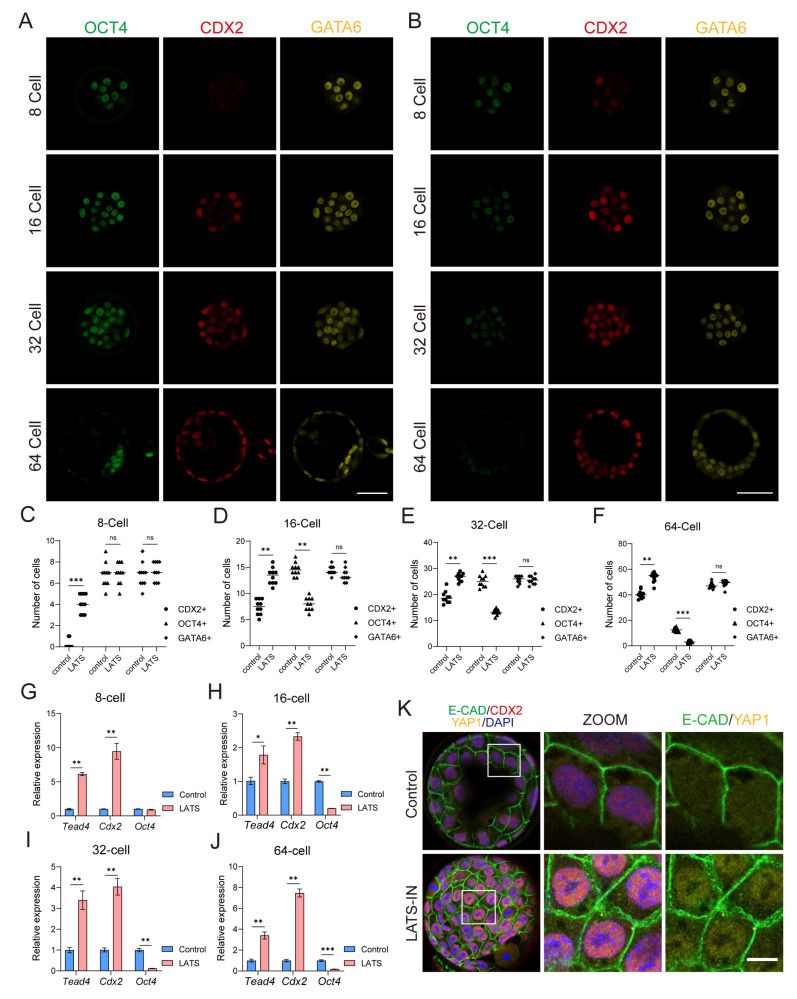
The cellular localization of YAP regulates the specialization of embryonic lineages. (**A**) Immunostaining for OCT4, CDX2, and GATA6 of the 8-cell, 16-cell, 32-cell, and 64-cell stage mouse embryos. (**B**) Immunostaining for OCT4, CDX2, and GATA6 of mouse embryos at different stages after LATS-IN-1 treatment. LATS-IN-1 factor at a concentration of 5 µM was added to the embryo culture medium from the 2-cell stage. (**C**–**F**) Comparison of the number of CDX2-positive, OCT4-positive, and GATA6-positive cells of the 8-cell stage, 16-cell stage, 32-cell stage, and 64-cell stage mouse embryos cultured with LATS-IN-1 factor. *n* = 10. (**G**–**J**) Relative expression of *Tead4*, *Cdx2*, and *Oct4* in the 8-cell, 16-cell, 32-cell, and 64-cell stage mouse embryos cultured with LATS-IN-1 factor were detected by qRT-PCR. qRT-PCR data were normalized to *Gapdh*, *n* = 3. (**K**) Immunostaining for E-CAD, CDX2, and YAP1 of the blastocyst (control) and LATS-IN-1 induced blastocysts. The enlarged region is indicated by the white box. The images of embryos were single focal plane, and the thickness of the optical section is 1.5 μm. For each graph, the data were represented as mean ± SEM, analyzed by Student’s *t*-test, ^ns^
*p* > 0.05, * *p* < 0.05, ** *p* < 0.01, *** *p* < 0.001. Scale bars: 50 μm in (**A**,**B**); 10 μm in (**K**).

**Figure 3 biology-13-00071-f003:**
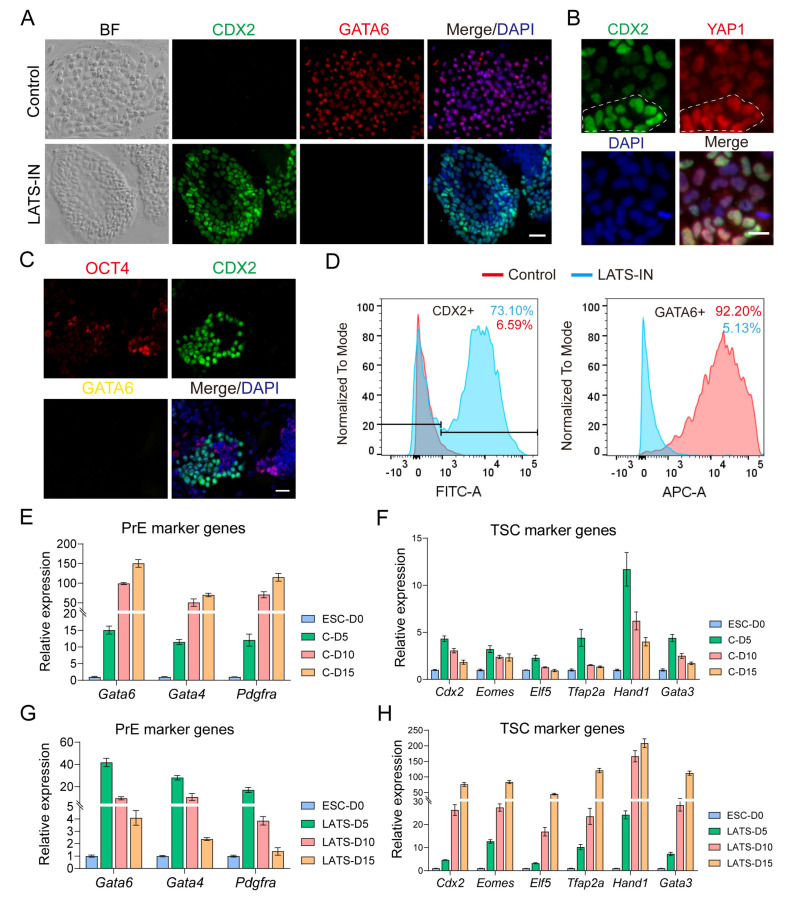
Inhibition of lats kinase can reprogram ESCs into TSC-like cells. (**A**) Immunostaining for CDX2 and GATA6 of ESC cultured with TSC medium or LATS-IN-1 induction medium for 15 days. Cell nuclei were stained with DAPI (blue). (**B**) Immunostaining for CDX2 and YAP1 of ESC cultured with LATS-IN-1 induction medium for 15 days. Cell nuclei were stained with DAPI (blue). The white dashed line indicates that the level of CDX2 expression is correlated with the amount of YAP1 accumulation in the nucleus. (**C**) Immunostaining for CDX2 and OCT4 of ESC cultured with LATS-IN-1 induction medium for 8 days. Cell nuclei were stained with DAPI (blue). (**D**) Flow cytometry analysis for CDX2 and GATA6 of ESC cultured with TSC medium or LATS-IN-1LATS-IN-1 induction medium for 10 days. (**E**) Relative expression of *Gata6*, *Gata4*, and *Pdgfra* in ESC cultured with TSC medium for 0, 5, 10, and 15 days. (**F**) Relative expression of *Cdx2*, *Eomes*, *Elf5*, *Tfap2c*, *Hand1*, and *Gata3* in ESC cultured with TSC medium for 0, 5, 10, and 15 days. (**G**) Relative expression of *Gata6*, *Gata4*, and *Pdgfra* in ESC cultured with LATS-IN-1 induction medium for 0, 5, 10, and 15 days. (**H**) Relative expression of *Cdx2*, *Eomes*, *Elf5*, *Tfap2c*, *Hand1*, and *Gata3* in ESC cultured with LATS-IN-1 induction medium for 0, 5, 10, and 15 days. All gene relative expression was detected by qRT-PCR, and the data were normalized to *Gapdh*, *n* = 3. For each graph, the data were represented as mean ± SEM. Scale bars: 20 μm in (**A**,**C**); 10 μm in (**B**).

**Figure 4 biology-13-00071-f004:**
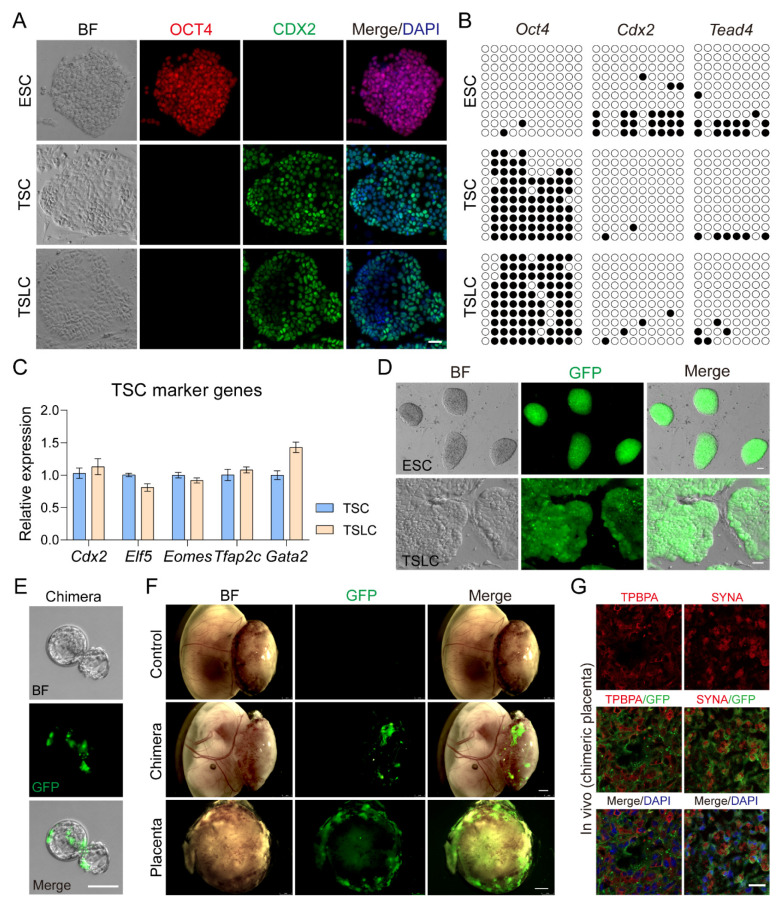
TSLC has the potential to differentiate into mature trophoblast cells in vivo. (**A**) Immunostaining for CDX2 and OCT4 of ESC, TSC, and TSLC. Cell nuclei were stained with DAPI (blue). (**B**) DNA methylation status of *Oct4*, *Cdx2*, and *Tead4* promoter regions in ESC, TSC, and TSLC. For each gene, Bisulfite sequencing of 10 samples was performed. Open and closed circles indicated the unmethylated and methylated CpGs. (**C**) Relative expression of *Cdx2*, *Eomes*, *Elf5*, *Tfap2c*, and *Gata2* in TSC and TSLC were detected by qRT-PCR. qRT-PCR data were normalized to *Gapdh*, *n* = 3. (**D**) Cell morphology of ESC (GFP labeled) and TSLC (GFP labeled). (**E**) Chimeric blastocyst formed after injection of 6–8 TSLCs (GFP labeled) into 32-cell stage embryos. (**F**) Mouse conceptuses of E13.5. TSLC (GFP labeled) contributed to the placenta formation. (**G**) Immunostaining for SYNA and TPBPA of the E13.5 chimeric placenta (GFP labeled). Cell nuclei were stained with DAPI (blue). For each graph, the data were represented as mean ± SEM. Scale bars: 20 μm in (**A**,**D**,**G**); 50 μm in (**E**); 1 mm in (**F**).

**Figure 5 biology-13-00071-f005:**
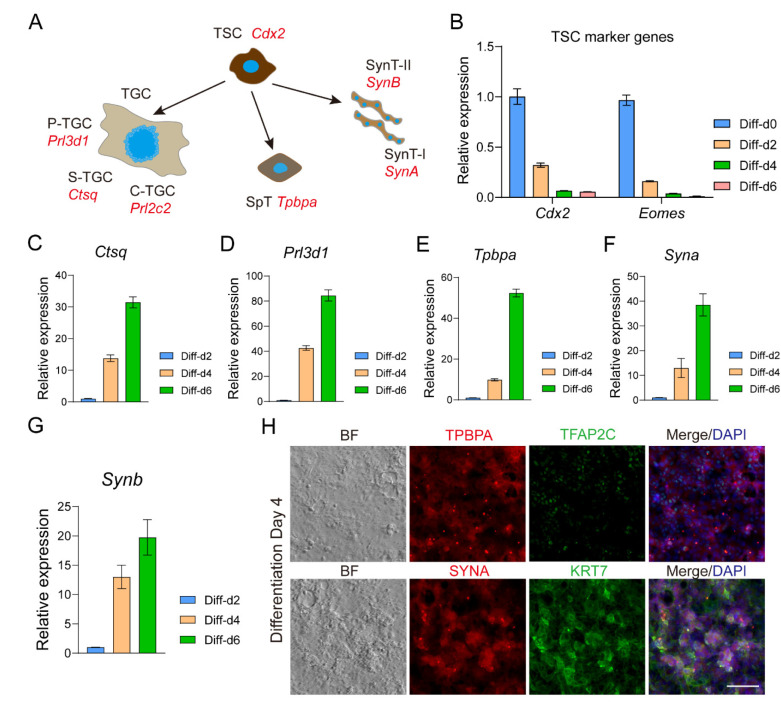
TSLC has the potential to differentiate into mature trophoblast cells in vitro. (**A**) Diagram showing the lineage of differentiation of trophoblast stem cells and their marker genes (red). TS, trophoblast stem cells; TGC, trophoblast giant cell; P-TGC, parietal TGC; S-TGC, sinusoidal TGC; SpA-TGC, spiral-associated TGC; C-TGC, canal TGC; SpT, spongiotrophoblast; SynT-I, syncytiotrophoblast layer I; SynT-II, syncytiotrophoblast layer II. (**B**) Relative expression of *Cdx2* and *Eomes* was detected by qRT-PCR during TSLC differentiation in vitro for 2, 4, and 6 days. qRT-PCR data were normalized to *Gapdh*, *n* = 3. (**C**–**G**) Relative expression for *Ctsq*, *Prl3d1*, *Tpbpa*, *Syna* and *Synb* was detected by qRT-PCR during TSLC differentiation in vitro for 2, 4, and 6 days. qRT-PCR data were normalized to *Gapdh*, *n* = 3. (**H**) Expression of *Krt7*, *Tfap2c*, *Tpbpa*, and *Syna* of TSLC differentiation for 4 days were detected by immunofluorescence. Cell nuclei were stained with DAPI (blue). For each graph, the data were represented as mean ± SEM. Scale bars: 50 μm in (**H**).

## Data Availability

Data are contained within the article and Appendix A.

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
