# Peer review of "Efficient Reprogramming of Mouse Embryonic Stem Cells into Trophoblast Stem-like Cells via Lats Kinase Inhibition"

_biology, 2024, doi:10.3390/biology13020071_

Round 1

Reviewer 1 Report

Comments and Suggestions for Authors

 In this work Gao et al. present LATS kinase inhibition as a method to promote trophoblast stem cell identity in vitro. They demonstrate the potential of using LATS kinase inhibition to generate trophoblast stem cell-like cultures from mouse ESC, something usually only achieved with overexpression with transcription factors such as CDX2.  I expect this would be of significance to the field, allowing for convenient generation of TSLCs in vitro.

The results are well presented overall, with mostly clear immunostaining imaging, and qPCR data with sufficient analyzed genes. The authors also strengthen their results by generating chimeric blastocysts, which is commendable.

However, the presented data also raises questions and concerns that need to be addressed (discussed below).  

1)     Figure 1A and S1A: The authors claim that LATS inhibition increases CDX2 and YAP1 expression. However, only one picture of each condition with high magnification is shown, which is insufficient to make this conclusion.  The two pictures appear to be of areas with different cell densities which may not be representative. The authors are requested to include lower magnification pictures (in supplementary data), and to perform quantification of CDX2 and YAP1 fluorescence intensity in at least a 100 nuclei per condition, and preferably with multiple replicates.

2)     Figure 1D and E: The finding that LATS inhibition would be capable of completely reversing the differentiation of TSC is remarkable and requires further substantiation. From the shown images it appears that this process is almost 100% efficient with all cells becoming CDX2 positive. To shed light on this, the authors are requested to include lower magnification pictures (in supplementary data).  In addition the authors are requested to characterize the dynamics of this process in more detail by performing a time course experiment where the expression of CDX2/YAP1 is examined at regular intervals during LATS inhibition (e.g. 24h, 48h and 72h). A control condition should be included where LATS inhibition is omitted (+FGF4/-LATS inhibition).

3)     Figure 2: the control and LATS+ embryos are imaged differently. The control embryo is shown as cross-section with cavity, while the LATS+ embryo is either imaged at the bottom or as a deeper Z-projection. The authors are requested to provide pictures for each condition that are imaged with a more similar method.  In addition the method of imaging  and image processing needs to be added to the figure description or method section. Are these Z-projections? And if so, what is the section depth in um?

4)     Figure 3:  In this key figure, the authors convincingly show that the in their LATS+ condition, an enrichment of CDX2+ cells occurs.  However, it leaves the reader with questions on how exactly this happens. The current presentation of the data suggests that reprogramming is a gradual process occurring over 15 days, with full reprogramming achieved at day 15. However, in supplementary figure 3C it is shown that at D5, CDX2+ cells are present as a small subpopulation in a mostly CDX2- culture. This would suggest that the actual generation of CDX2+ cells  already occurs between D0 and D5, and that between D5-D15, the +LATS culture selects for these cells over time.  This is a very relevant distinction, especially because the authors also show that induction of TSC marker genes (Figure 3F and H) is very similar between control and +LATS. This raises the question if induction of CDX2+ cells in the D0-D5 period is not actually dependent on +LATS, but that the +LATS condition subsequently promotes survival and proliferation of the TSCLC.

a)      Can the authors comment on this interpretation of the data?  

b)     A more detailed characterization of the D0-D5 induction period is needed. The authors are requested to image CDX2/KRT7 positive cells at 24h intervals from D0-D5, in both control (TSC-medium) and +LATS.  This should be accompanied by a quantification of the amount of CDX2+ cells.  This will provide the reader with clearer information about when these CDX2+ cells occur, and what the difference is between control and +LATS in the D0-D5 time period.

c)      The corresponding description in the methods section (line 171-176) is not detailed enough for other researchers to reproduce the authors. Specifically, the authors are unclear about how they perform passaging during the reprogramming protocol, including:  what detachment method is used, what split ratio or cell count is used, and how often is passaged during the 15 day period.

d)     Figure 3C is not clearly discussed in the text.  It is referenced in line 333 as a part of a sentence on GATA expression, while no GATA6 is shown in the images in 3C. In addition, why is day 8 differentiation shown here, while this is not timepoint analyzed in any of the other figures?

e)     Figure 3B: The authors are requested to show a picture with lower magnification as they did in figure 3A. With the limited amount of cells shown here it is difficult to validate their conclusion that CDX2 and YAP1 expression are correlated.

5)     Figure 4G: It is difficult to differentiate GFP positive and endogenous GFP negative cells in the provided images. Are all the cells in the image GFP positive? The authors are requested to either: [1] include a negative control for the GFP staining (in control placenta), or [2] change the current images to images of the tissue that clearly contain both injected GFP+ cells and endogenous GFP- cells that serve as an inherent negative control.

6)     Figure 5H:  It is difficult to assess whether the immunostainings of TPBPA and SYNA are specific. The authors are requested to include a negative control (undifferentiated TSLC at Day 0).

7)     Line 178:  This appears to be an answer from a large language AI model that the authors left in.

8)     Section 3.5 (lines 407-444):  The entire appears to be duplicated, with lines 407-425 being a slightly altered version of 426-444.

Reviewer 2 Report

Comments and Suggestions for Authors

The study focused on the transition from embryonic stem cells ESC to trophoblast stem-like cells TSLC in mouse embryos, hindered by distinct lineage differences.  Inhibiting LATS kinase facilitated the conversion of inner cell mass to trophectoderm and transformed ESC into stable self-renewing TSLC.  The TSLC exhibited distinct molecular properties, expressing marker genes like Cdx2 and Eomes similar to trophoblast stem cells TSC.  Moreover, TSLC demonstrated the ability to differentiate into mature trophoblast cells in vitro and actively participated in placenta formation in vivo, emphasizing their role in cell fate determination. 

1.       Graphic abstract is misleading for readers to understand the mechanism of LATS-YAP axis function.  The notation "LATS(off)" is ambiguous and makes it unclear whether it implies the activation of LATS suppress YAP or the inactivation of LATS suppressing YAP.  In the graphic abstract, it should be described as 'LATS' without 'off' to avoid confusion.  Also, the samples undergoing LATS inhibition were labeled as +LATS; however, this notation may be misleading as it initially appears as if LATS is being added or activated. Therefore, the notation at this point should also be revised.

2.       The authors claim that, in the control group, YAP1 mainly accumulates in the cytoplasm in Fig. 2K, however, it appears that YAP1 is localized in both the cytoplasm and the nucleus.

3.       In Fig. 3D: Why did two peaks appear in the FACS analysis of CDX2 expression in samples treated with LATS inhibitor?

4.       In mammals, there are two forms of LATS, LATS1 and LATS2.  In the context of differentiation from embryonic stem cells to trophoblast stem-(like) cells, which of these LATS forms is considered to play a more dominant role?

Reviewer 3 Report

Comments and Suggestions for Authors

The authors present a novel method of derivation of mouse trophoblast-stem-like cells directly from ES cells via LATS kinase inhibition, and provide data on LATS inhibition effects in intact mouse embryos. This is in contrast to standard TS-cell derivation protocol, which does not enable ES cell reprogramming. The findings can be therefore considered novel and of interest for both practical aspects of stem cell research and in vitro modeling of development, and shed more light on LATS kinase role in TE differentiation in vivo. In my opinion the claims are supported and the trophoblast properties of newly derived cell line have been satisfactorily proven.

It is not entirely clear whether cells derived with this new protocol should be considered bona fide TS cells, or have different enough properties to be considered a different population. Authors provide data to support the former, yet are aware further studies are necessary. However, there seems to be some confusion throughout the manuscript as to which nomenclature is used, which needs to be revised. Some other aspects of the manuscript also need to be revised, to improve the quality, which are listed below:

The main concern is the use of embryo culture media dedicated to human preimplantation embryo (and containing HSA), rather than protocols for mouse embryo culture. While the embryos presented look healthy, and the treatment poses no concern for cell line portion of the work, different media composition can slightly affect lineage differentiation (consult Frum and Ralston, 2020). Why was this protocol chosen? It should be discussed, and how it might affect the outcome.

Please revise manuscript for proper gene/protein symbol nomenclature (mouse gene symbols should be italicized lower case, protein symbols not italicized upper case; human gene symbols – italicized upper case etc.) – there are some inconsistencies.

Please revise in vivo and in vitro italicization, it’s inconsistent.

Literature citations – overall high quality studies relevant to the topic of the paper are cited, but sometimes citations seem to be misplaced. For example line 45 – citations 1&4 report specific findings related to hippo &notch pathways, while earlier/more general findings would be more relevant, l. 45-57 – mostly citations about stem cell lines instead of formation of early cell lineages, l. 64 also not exactly placed. L.71, (and 26. In l.125) – these citations relate to LATS and probably more suitable for the following sentences.

I also strongly recommend citing some original studies in relation to cell polarity in morula, compaction etc., original paper by Rivron et al., 2018 in relation to blastoids and van den Brink et al., 2014 for gastruloids (88-91), an in relation to hippo pathway Hirate et al., 2012, needs to be also cited (in addition to, or instead, Home et al)

Introduction l. 120-121 “independent expression” – I understand the intent, but if mutual inhibition is at play, then they are rather dependent on each other.

TSCs vs TSLCs

Both TSCs derived with previously established protocol, and novel TSLCs are described in the study, but the nomenclature seems to inconsistent.

l.32, 34 – are these TSCs or TSLCs?

Images of the embryos –are these 3D reconstruction, single focal plane, extended focus? What was the thickness of the optical section. Please provide information and present in consistent manner.

Mouse and cell lines

l.138 – which mouse line was used in which experiment (mention here or in results section). What were EGFP mice used for? Esc derivation?

l. 154 – is B6-G/R line same as CAG-EGFP mentioned above? Not clear.

l. 164 – this passage reads as if the authors received a single blastocyst from dr Rossant laboratory, rather than the cell line. Was that the case? If not, please revise the text, and also provide a reference for an original paper from Rossant lab, if applicable.

l. 386 ESCs-ZsGreen – this line is not mentioned elsewhere, should be included in Materials section with proper references

Methods section – l. 166 – reference for TSC protocol needed

Whole section 2.3  The sequence of action not entirely clear, please consider revising.

l. 181, 204 – do you mean “paraformaldehyde”?

l. 216 – trophoblast or trophoblastic?

Results

235-237 –describe the outcome of validation

Section 3.2 – following LATS inhibition, are all cells at 32- and 64- cell stage CDX2- positive, both inside and outside? Providing percentage – lineage composition – of embryos with and without treatment would be more useful. Are there cells co-expressing OCT4 and CDX2, and how the treatment changes their contribution in TE and ICM? Please discuss the identity of CDX2-positive cells in the ICM (or inside compartment) of the blastocyst.

Section 3.3, l. 323 and following – would these cells be described as TSCs or TSLCs?

Section 3.4 – How are these cells purified, and for how many passages can they be maintained without change of properties?

In section 3.5 there is a duplication of text (408-425 vs 426-444)

l. 178-179 – remove?

I.504-507 – unclear, please review sentence structure

Discussion

l. 463-464 –please revise the sentence, unclear

l. 465-467 – While Posfai et al. publication indeed shows very limited TE potential of EPSCs, 2C-like cells and MSCs have been shown to differentiate towards TE. As these finding present protocols of somewhat similar practical implications to TSLC presented in current manuscript, please revise this section and discuss correctly and in more detail (how is your approach similar/better in relation to specification towards TE in vitrog).

513- I recommend removal of “not shown” data info, does not seem to be necessary

303-3-5 There is indeed accumulation of YAP1 in the nuclei of LATS inhibitor treated embryos, but the control group seems to have YAP1in both cytoplasm and nucleus, not just cytoplasm.

484- mural vs polar trophectoderm. If the authors indeed observed differences between YAP1 mural and polar TE localization and want to draw conclusions from this observations, this should be described in in results section in more detail (percentage of cells with nuclear accumulation of YAP?). Earlier studies (for example, fig. 1 D in Hirate et al., 2012, or 1G in Hashimito & Sasaki, 2019 (and following figures)) do not show such difference, so this claim should be very well supported or removed.

Round 2

Reviewer 1 Report

Comments and Suggestions for Authors

The authors have sufficiently  addressed my comments.